# Release of Sulfur and Chlorine Gas Species during Combustion and Pyrolysis of Walnut Shells in an Entrained Flow Reactor

Coskun Yildiz *[ID], Marcel Richter [ID], Jochen Ströhle [ID] and Bernd Epple [ID]

Institute for Energy Systems and Technology (EST), Technical University of Darmstadt, Otto-Berndt-Straße 2, 64287 Darmstadt, Germany; marcel.richter@est.tu-darmstadt.de (M.R.); jochen.stroehle@est.tu-darmstadt.de (J.S.); bernd.epple@est.tu-darmstadt.de (B.E.)
* Correspondence: coskun.yildiz@wihi.tu-darmstadt.de

**Abstract:** The release behavior of sulfur and chlorine compounds into the gas phase of walnut shell particles (WNS) is studied with an entrained flow reactor. Experiments are carried out in nitrogen ($N_2$), carbon dioxide ($CO_2$) atmosphere and under air and oxy-fuel conditions at different temperatures ($T$ = 1000–1300 °C) and stoichiometries ($\lambda$ = 0.8–1.1). A total of 98.7% of fuel-bound sulfur volatilizes as sulfur dioxide ($SO_2$), carbonyl sulfide (COS) and hydrogen sulfide ($H_2S$) in the gas phase in $N_2$ atmosphere at 1000 °C. As hydrogen chloride (HCl), 37.0% of the chlorine is released at this temperature. In $CO_2$ atmosphere, a similar total release of sulfur and chlorine is observed (1000 °C). With each temperature increment, the release of $SO_2$, $H_2S$ and HCl in the gas phase decreases ($N_2$ and $CO_2$ atmosphere). $SO_2$ forms the major sulfur component in both atmospheres. In $CO_2$ atmosphere, higher concentrations of COS were detected than in $N_2$ atmosphere. Air and oxy-fuel combustion conditions show significantly lower $SO_2$, COS and HCl concentrations as in $N_2$ and $CO_2$ atmosphere. No $H_2S$ is detected in the gas phase during any of the combustion trials.

**Keywords:** walnut shell; entrained flow reactor; pyrolysis; gasification; combustion; oxy-fuel; sulfur; chlorine





## 1. Introduction

In order to limit the global warming target to 1.5 °C, as set out in the Paris Agreement [1], carbon dioxide emissions must be reduced to zero by about 2055 [2]. Currently, about 79% of global carbon dioxide ($CO_2$) emissions are energy related [3]. The largest contribution to reducing these emissions can come from the energy sector [2]. Current projections for primary energy demand do not foresee a decrease in absolute fossil fuel consumption until 2040 [3]. Therefore, a significant part of the solutions to reduce $CO_2$ emissions will be the application of carbon capture, utilization and storage (CCUS) technologies.

One advanced method for achieving CCUS in combustion systems is the oxy-fuel process. In this process, the atmosphere in the combustion zone consists mainly of carbon dioxide, water ($H_2O$) and oxygen ($O_2$). The major advantage is the ability to produce a flue gas that consists almost entirely of $CO_2$, which facilitates utilization or storage.

Combining the oxy-fuel process with CCUS allows for nearly carbon neutral fossil fuel combustion. If biomass is used instead of fossil fuels, a net carbon sink is created, which results in net negative $CO_2$ emissions [2,4]. The process is referred to as bioenergy carbon capture, utilization and storage (BECCUS), and is expected to account for a relevant share of future primary energy supply [2,5]. However, the replacement of conventional fossil fuels with biomass is associated with a number of barriers that need to be resolved. The main problem is the strong formation of deposits that promote corrosion when firing biomass fuels, as they are particularly rich in sulfur (S), chlorine (Cl) and potassium (K) [6,7]. The formation of the gases ($SO_2$, COS, $H_2S$ and HCl) is critical for several reasons. The concentrations in the flue gas may not exceed the country-specific emission control

limits. This can require costly measures in power plants, such as flue gas desulfurization. On the other hand, the gases promote corrosion in power plants. This may cause a direct corrosion by accelerating the oxidation of the metal alloys [8,9], or the interaction with alkali metals contained in biomass. Alkali sulfates (e.g., sodium sulfate ($NaSO_4$), dipotassium sulfate ($K_2SO_4$)) or alkali chlorides (e.g., sodium chloride (NaCl), potassium chloride (KCl)) can be formed. These compounds form deposits on the heating surfaces and are extremely corrosive [10,11]. Due to the flue gas recirculation during oxy-fuel combustion, corrosive gases and deposits can accumulate in the combustion chamber. This can result in a high corrosion in oxy-fuel power plants.

The formation of sulfur in biomass is divided into organic and inorganic forms [12]. The organic part is released at low temperatures. It is assumed that the release of sulfur in biomass starts at about 180 °C. The decomposition of cysteine and methionine, the two most important sulfur-containing precursors of plant proteins, takes place at 178 and 183 °C [6,13]. The inorganic fraction remains in the char during pyrolysis and is released at temperatures above 900 °C [6,14]. It is observed that the amount of sulfur released into the gas phase during the pyrolysis of biomass at temperatures of 400 °C is more than 50%. While at temperatures above 500 °C, only a small amount of additional sulfur is released [15]. Released sulfur compounds can be reabsorbed by pyrolysis temperatures of more than 325 °C [15]. Thermodynamic equilibrium calculations indicate that sulfatic sulfur is transformed by decomposition and interactions with organic material [16]. The second release step occurs during the burnout of the char [12]. The release pathways include the evaporation of alkali sulfates at temperatures above 1000 °C or the decomposition of sulfate, releasing $SO_2$ into the gas phase. The balance between organic and inorganic sulfur species is important for the release behavior. The two sulfur species are mainly relevant in different temperature ranges and thus in different stages of the combustion process. Released sulfur can be reabsorbed into the fuel bed through secondary reactions with the char matrix. Subsequently, it is released again during char conversion along with the high-temperature inorganic release of S compounds [17].

The release of chlorine gas species is an important factor in thermal conversion of fuels. On the one hand, chlorine gas species are air pollutants [18], and on the other hand, chlorine release promotes the forming of corrosive deposits in power plants [19,20]. Chlorine species released from the combustion of biomass are the main cause of corrosion in grid firing [21,22]. Chlorine bound in solid fuels is classified into inorganic and organic chlorine. Inorganic chlorine consists of mainly chlorides of sodium or potassium [23]. Studies show that chlorine release occurs from both organic and inorganic fraction from fuel [14,24,25]. Chlorine, which is released at low temperatures in the form of HCl [14,25] can potentially be recovered by secondary reactions with available metals in the fuel [14,26]. Studies show KCl is the main chlorine species in biomass [14]. Thermodynamic calculations show the preferential bonding between chlorine and potassium in the solid and gaseous phases [22,27]. Experimental and simulative studies show that sulfation of KCl releases HCl [6,28].

The approach of this study is to convert existing power plants to oxy-fuel combustion by flue gas recirculation. Using walnut shells as renewable can create net negative $CO_2$ emissions during electricity generation. However, the sulfur and chlorine release during walnut shell oxy-fuel combustion is poorly studied. The change in the combustion atmosphere from $O_2/N_2$ to $O_2/CO_2$ affects the release behavior. This can have negative effects on power plants, such as corrosion or pollutant formation. In order to prevent corrosion and pollutant formation, the sulfur and chlorine release is investigated in this study. The experiments are carried out in an entrained flow reactor with particle heating rates comparable to large-scale furnaces. We provide experimental data necessary for further the development of a sulfur and chlorine reaction mechanism. The release is studied in four atmospheres: $N_2$ and $CO_2$ to describe the early combustion process in which walnut shell particles devolatilize, and $O_2/N_2$ and $O_2/CO_2$ to investigate the influence of char conversion.

## 2. Material and Methods

### 2.1. Fuel Characterization

Pulverized walnut shell particles with a size range of 100–250 µm were used for the investigations in this study. Moisture, volatile and ash content of the fuel are determined according to the German standards [29–31]. Sulfur and chlorine content are determined according to the German standards DIN 51724 and DIN 51727 [32,33]. The sulfur and chlorine content amounts to 0.04% and 0.29%, respectively. Ultimate, proximate analysis and ash composition are summarized in Table 1.

**Table 1.** Chemical properties of walnut shell particles.

| | | | |
|---|---|---|---|
| C | [wt%] | daf [a] | 48.73 |
| N | [wt%] | daf | 0.15 |
| H | [wt%] | daf | 6.30 |
| S | [wt%] | daf | 0.04 |
| Cl | [wt%] | daf | 0.29 |
| O [b] | [wt%] | daf | 35.1 |
| Ash | [wt%] | dry | 0.64 |
| Water | [wt%] | aa [c] | 6.00 |
| Volatiles | [wt%] | daf | 81.07 |
| High heating value | [MJ kg$^{-1}$] | dry | 20.51 |
| Al | [wt%] | ash | 0.16 |
| Ca | [wt%] | ash | 7.93 |
| Fe | [wt%] | ash | 0.21 |
| K | [wt%] | ash | 29.14 |
| Mg | [wt%] | ash | 1.33 |
| Na | [wt%] | ash | 1.63 |
| P | [wt%] | ash | 1.41 |
| Si | [wt%] | ash | 1.01 |

[a] reference state: dry, ash-free; [b] by difference; [c] reference state: as analyzed.

### 2.2. Entrained Flow Reactor

An entrained flow reactor (EFR) is used to carry out the experiments. Details are shown in Figure 1. It is heated electrically (up to 1600 °C) and can be operated under pressure (up to 20 bar). Nitrogen, carbon dioxide, air and oxygen (or a mixture thereof) can be supplied to the reactor. Pulverized fuel is fed from the top in the reaction zone (feed rates up to 500 g/h). The dosing system is located above the reactor (not shown in Figure 1). A pressure vessel surrounds the system. A constant gas flow (co-flow) is steaming through the dosing system into the particle injection lance. A container, which is continuously stirred, contains the fuel particles. Fuel particles are gravimetrically dosed via a screw feeder and fall from the outlet of the screw into a vibrating trough, which homogenizes the particle stream. The container and screw feeder are mounted on load cells. This allows controlled operation of the screw and dosing with low fluctuations (approximately ±5 g/h around the mean value). The particles are transported into the reaction zone through a water-cooled injection lance. Due to the water cooling, the temperature of the particle-loaded carrier gas stream is less than 20 °C. This prevents premature thermal reaction of the fuel particles. The lance is located in the preheating zone of the reactor. This consists of nine electrically heated secondary gas flow paths (co-flow), which heat the co-flow to the temperature of the reaction zone. A fixed bed of aluminum oxide spheres is located in gas flow paths to improve heat transfer. At the bottom part of the preheating section is an injection nozzle located where the particle-loaded primary gas flow and co-flow enter the reaction zone. The co-flow enters the reaction zone through an annular gap. By preheating the co-flow, particle heating rates of up to $10^4$ K/s can be achieved, which are comparable to heating rates of a large-scale furnace.

The main component of the reactor is a vertical reaction zone made of ceramic tube with a total length of 2200 mm and an inner diameter of 70 mm. The wall temperature of the reaction zone is regulated by six heating elements on each level installed around the reaction zone. Gas samples can be collected at different positions in the reaction zone (gas sampling ports 3–5). The distance from the injector nozzle amounts to 600, 1300 and 2000 mm, respectively.

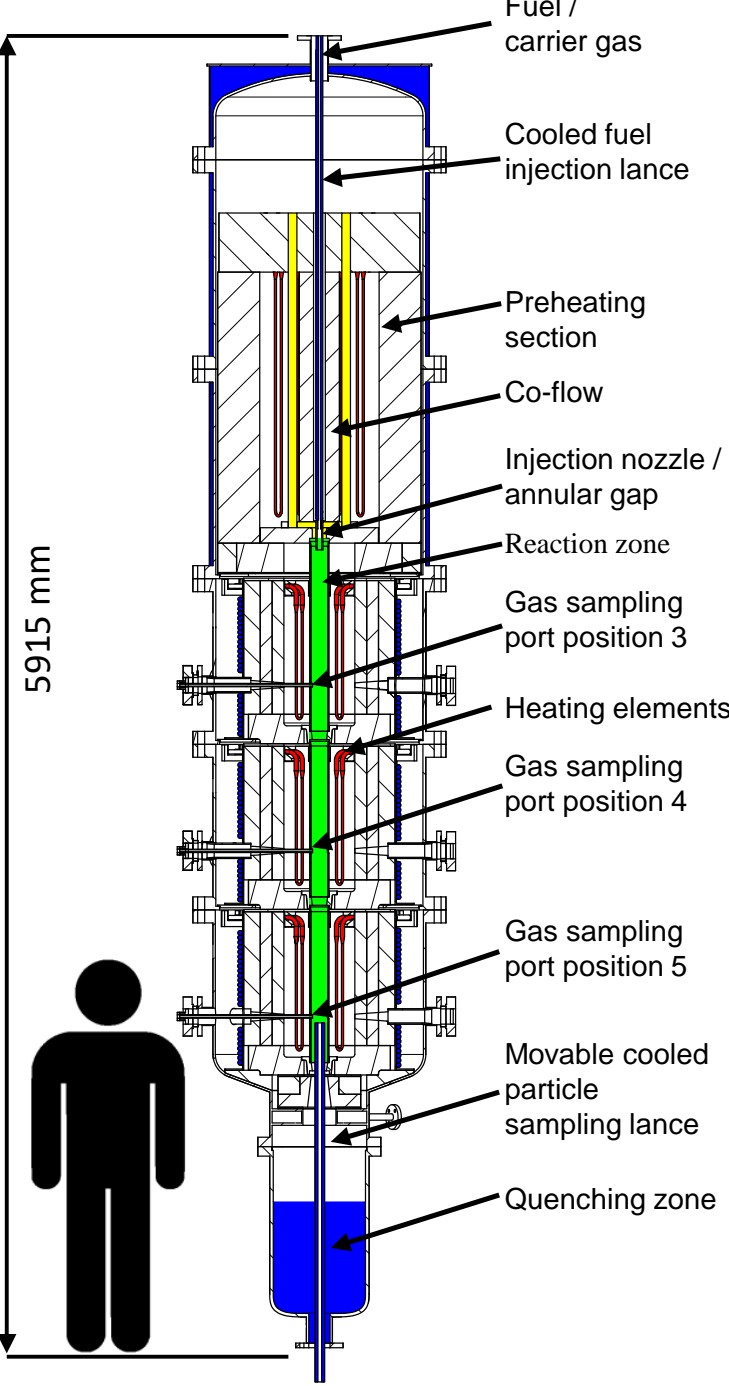

**Figure 1.** Drawing of the entrained flow reactor.

### 2.3. Measuring Techniques

Gas samples are taken at position 4 (Figure 1) with a ceramic tube. The mean particle residence time is estimated to be 0.5 s according to [34]. The ceramic tube enters the reaction zone of the EFR vertically. A gas flow of 40 L/h is extracted from the reaction zone. A PTFE fine filter is positioned directly after the ceramic tube, which removes the majority of particulate matter from the sampling gas. The sampling gas is transported through a 1 m long flexible PTFE tube with an inner diameter of 4 mm to a mass spectrometer and gas suction pump. A filter, tubes and all other parts in contact with the sampling gas after the outlet of the reactor were heated up to 180 °C to prevent condensation in the sampling line. The inlet of the mass spectrometer (GAM 200, InProcess Instruments, Bremen, Germany) is positioned in a T-connector located in the sampling line. The spectrometer is calibrated for the following 10 gas species: nitrogen ($N_2$), oxygen ($O_2$), argon (Ar), carbon monoxide (CO), carbon dioxide ($CO_2$), methane ($CH_4$), sulfur dioxide ($SO_2$), carbonyl sulfide (COS), hydrogen sulfide ($H_2S$) and hydrogen chlorine (HCl). In addition, ionic currents with mass-to-charge ratios (m/z) of 47 amu and 76 amu are recorded. These are proportional to carbon disulfide ($CS_2$) and methanethiol ($CH_4S$) concentrations.

### 2.4. Determination of Sulfur and Chlorine Release

For the discussion of the EFR experiments, it is of interest to determine the fraction of sulfur or chlorine release from the fuel. For this purpose, the molar flue gas flow $\dot{n}_{FG}$ is calculated. It is assumed $N_2$ is not involved in the reactions. Thus, the molar mass flow rate of $N_2$ entering the reactor corresponds to the molar mass flow rate present in the flue gas. The molar mass flow rate of the flue gas ($\dot{n}_{FG}$) is determined via a $N_2$ balance according to the following equations by using the constant $N_2$ molar mass flow rate ($\dot{n}_{N_2}$) and the measured volume fraction ($x_{N_2}$):

$$\dot{n}_{FG} = \frac{x_{N_2}^0}{x_{N_2}} \dot{n}_{N_2} = \frac{x_{N_2}^0}{x_{N_2}} \frac{p \dot{V}_{N_2}}{\overline{R} T} \tag{1}$$

Equation (1) contains the nitrogen fraction ($x_{N_2}^0$) of the gas entering the reactor, the pressure $p$, the volume flow ($\dot{V}$), the temperature ($T$) and the universal gas constant ($R$).

The yields of the specific sulfur species ($y_{SO_2}$, $y_{H_2S}$, $y_{COS}$) are defined as the ratio of sulfur present in the gas phase as $SO_2$, COS and $H_2S$, ($\dot{n}_{SO_2}$, $\dot{n}_{COS}$, $\dot{n}_{H_2S}$) and the amount of sulfur entering the reactor ($\dot{n}_{S, Fuel}$) according to Equations (2)–(4) [35]:

$$y_{SO_2} = \frac{\dot{n}_{SO_2}}{\dot{n}_{S, Fuel}} = \dot{n}_{FG}\, x_{SO_2} \left( \frac{\dot{m}_{Fuel}\, w_{S, Fuel}}{M_S} \right)^{-1} \tag{2}$$

$$y_{COS} = \frac{\dot{n}_{COS}}{\dot{n}_{S, Fuel}} = \dot{n}_{FG}\, x_{COS} \left( \frac{\dot{m}_{Fuel}\, w_{S, Fuel}}{M_S} \right)^{-1} \tag{3}$$

$$y_{H_2S} = \frac{\dot{n}_{H_2S}}{\dot{n}_{S, Fuel}} = \dot{n}_{FG}\, x_{H_2S} \left( \frac{\dot{m}_{Fuel}\, w_{S, Fuel}}{M_S} \right)^{-1} \tag{4}$$

The chlorine release ($y_{Cl}$) is defined as the ratio of the amount of substance chlorine present in the gas phase as HCl ($\dot{n}_{HCl}$) and the amount of chlorine ($\dot{n}_{Cl}$) entering the reactor, according to Equation (5):

$$y_{HCl} = \frac{\dot{n}_{HCl}}{\dot{n}_{Cl, Fuel}} = \dot{n}_{FG}\, x_{HCl} \left( \frac{\dot{m}_{Fuel}\, w_{Cl, Fuel}}{M_{Cl}} \right)^{-1} \tag{5}$$

In Equations (2)–(5), the molar mass flow rate of the flue gas ($\dot{n}_{FG}$), the measured molar mass fraction of sulfur species ($x_{SO_2}$, $x_{H_2S}$, $x_{COS}$), the molar mass fraction of HCl ($x_{HCl}$), the fuel mass flow rate ($\dot{m}_{Fuel}$), the sulfur and chlorine content of the fuel ($w_S$) and ($w_{Cl}$) and the molar masses of sulfur and chlorine ($M_S$ and $M_{Cl}$) are used for the calculations.

The total sulfur release ($y_S$) is calculated from the sum of the release of the individual species (Equation (6)):

$$y_S = y_{SO_2} + y_{COS} + y_{H_2S} \tag{6}$$

During the experiments in $CO_2$ and $O_2/CO_2$ atmospheres, no more inert gas ($N_2$) is introduced into the reactor. The determination of the flue gas flow according to Equations (2)–(5) is not possible. It is assumed that the molar mass flow rate of the flue gas $\dot{n}_{FG}$ in $CO_2$ atmosphere is equal to the molar mass flow rate in $N_2$ atmosphere. This assumption is made because the gas flow entering the reactor is 20 times higher than the fuel flow. The error amounts a maximum of 1% if the fuel is completely converted in the $CO_2$ atmosphere.

Further, it is assumed that the molar mass flow rate of the flue gas in air atmosphere corresponds to the oxy-fuel conditions. Even due to the high $CO_2$ content in the $O_2/CO_2$ atmosphere, the char conversion could be higher than in air atmosphere. The maximum error amounts to 1% for a sub-stoichiometric combustion.

For the calculation of the sulfur or chlorine release, quantities with uncertainties are used (Table 2). By the applied calculation operations (Equations (1)–(6)), these uncertainties are combined according to DIN 1319-4 [36].

**Table 2.** Uncertainties of the quantities used in the determination of sulfur and chlorine release.

| Symbol | Uncertainty [a] | Note |
|---|---|---|
| $\dot{V}$ | max. 0.4% | Gas flow measurement |
| $x_{N_2}$ | $\sigma$ | MS measurement |
| $\dot{m}_{Fuel}$ | $\sigma$ | MFC measurement |
| $w_{S, Fuel}$ | 3% MV | Standard of certified laboratory |
| $w_{Cl, Fuel}$ | 3% MV | Standard of certified laboratory |
| $x_{SO_2}$ | $\sigma$ | MS measurement |
| $x_{H_2S}$ | $\sigma$ | MS measurement |
| $x_{COS}$ | $\sigma$ | MS measurement |
| $x_{HCl}$ | $\sigma$ | MS measurement |

[a] $\sigma$: standard deviation; MV: measured value.

### 2.5. Experimental Procedure

Sulfur and chlorine concentrations are determined in various atmospheres ($N_2$, $CO_2$, $O_2/N_2$ and $O_2/CO_2$) and different temperatures (1000–1300 °C). In combustion experiments (air and oxy-fuel conditions), the stoichiometric ratios ($0.8 < \lambda < 1.1$) are additionally varied. The gas sampling is performed at position 4 with a particle residence time of approximately 0.5 s. The distance to the injector nozzle amounts to 1300 mm (Figure 1).

The Experiments in $N_2$ and $CO_2$ atmosphere are performed with a constant fuel mass flow ($\dot{m}_{Fuel}$) of 0.15 kgh$^{-1}$ and a constant gas flow ($\dot{V}$) of 2.5 normal m$^3$h$^{-1}$. The experiments in $O_2/N_2$ and $O_2/CO_2$ atmosphere are performed with a constant fuel mass flow of 0.2 kgh$^{-1}$ and a gas flow, which varies depending on the stoichiometric ratio (1 m$^3$h$^{-1} < \dot{V} < 1.38$ m$^3$h$^{-1}$). The EFR is operated at a maximum atmospheric over pressure of 3.5 mbar during the experiments.

For this study, a total of 24 different experiments are performed. The experimental parameters of fuel supply ($\dot{m}_{Fuel}$), gas flow ($\dot{V}$), oxygen-fuel equivalence ratio ($\lambda$) and temperature ($T$) are listed in Table 3. Prior to all experiments, the EFR and the fuel supply are purged with the specific gas for three hours to set the required atmosphere in the reaction zone. The atmosphere is controlled with the mass spectrometer. After the temperature of the reaction zone and co-flow is reached, the fuel feed is started. Once steady-state conditions (stable gas species concentrations) are reached, the gas concentrations are recorded for at least 10 min.

**Table 3.** Parameters of experiments.

| Trial Number | Atmosphere | $\dot{m}$ [kg h$^{-1}$] | $\dot{V}$ [Normal m$^3$h$^{-1}$] | $\lambda$ [-] | $T$ [°C] |
|---|---|---|---|---|---|
| 1 | 100% N$_2$ | 0.15 | 2.5 | - | 1000 |
| 2 | 100% N$_2$ | 0.15 | 2.5 | - | 1100 |
| 3 | 100% N$_2$ | 0.15 | 2.5 | - | 1200 |
| 4 | 100% N$_2$ | 0.15 | 2.5 | - | 1300 |
| 5 | 100% CO$_2$ | 0.15 | 2.5 | - | 1000 |
| 6 | 100% CO$_2$ | 0.15 | 2.5 | - | 1100 |
| 7 | 100% CO$_2$ | 0.15 | 2.5 | - | 1200 |
| 8 | 100% CO$_2$ | 0.15 | 2.5 | - | 1300 |
| 9 | 20.95% O$_2$/79.05% N$_2$ | 0.2 | 1.00 | 0.8 | 1000 |
| 10 | 20.95% O$_2$/79.05% N$_2$ | 0.2 | 1.13 | 0.9 | 1000 |
| 11 | 20.95% O$_2$/79.05% N$_2$ | 0.2 | 1.25 | 1.0 | 1000 |
| 12 | 20.95% O$_2$/79.05% N$_2$ | 0.2 | 1.38 | 1.1 | 1000 |
| 13 | 20.95% O$_2$/79.05% CO$_2$ | 0.2 | 1.00 | 0.8 | 1000 |
| 14 | 20.95% O$_2$/79.05% CO$_2$ | 0.2 | 1.13 | 0.9 | 1000 |
| 15 | 20.95% O$_2$/79.05% CO$_2$ | 0.2 | 1.25 | 1.0 | 1000 |
| 16 | 20.95% O$_2$/79.05% CO$_2$ | 0.2 | 1.38 | 1.1 | 1000 |
| 17 | 20.95% O$_2$/79.05% N$_2$ | 0.2 | 1.00 | 0.8 | 1300 |
| 18 | 20.95% O$_2$/79.05% N$_2$ | 0.2 | 1.13 | 0.9 | 1300 |
| 19 | 20.95% O$_2$/79.05% N$_2$ | 0.2 | 1.25 | 1.0 | 1300 |
| 20 | 20.95% O$_2$/79.05% N$_2$ | 0.2 | 1.38 | 1.1 | 1300 |
| 21 | 20.95% O$_2$/79.05% CO$_2$ | 0.2 | 1.00 | 0.8 | 1300 |
| 22 | 20.95% O$_2$/79.05% CO$_2$ | 0.2 | 1.13 | 0.9 | 1300 |
| 23 | 20.95% O$_2$/79.05% CO$_2$ | 0.2 | 1.25 | 1.0 | 1300 |
| 24 | 20.95% O$_2$/79.05% CO$_2$ | 0.2 | 1.38 | 1.1 | 1300 |

## 3. Results and Discussion

The EFR is operated under four different atmospheres: N$_2$ and CO$_2$ to study sulfur and chlorine release in the early stage of the combustion process, and combustion conditions (air and oxy-fuel) to study the influence of char conversion at different stoichiometric ratios. An overview of the experiments is given in Table 3.

### 3.1. Pyrolysis in Nitrogen and Gasification Carbon Dioxide Atmosphere

The release of sulfur and chlorine species (SO$_2$, H$_2$S, COS and HCl) during pyrolysis in nitrogen atmosphere is shown in Figure 2. All of the sulfur and chlorine species under consideration occur in the gas phase. The species have their maximum at 1000 °C and their concentrations decrease with increasing temperature. SO$_2$ is the major sulfur species and its maximum concentration amounts to 35 ppm. The release of COS is less than SO$_2$, the maximum concentration is 5 ppm at 1000 °C. H$_2$S is detected at low levels (<1 ppm). It is assumed H$_2$S is released during pyrolysis and reacts in the gas phase with CO or CO$_2$ to COS according to Equations (7) and (8) [37,38]. A high temperature favors this behavior, so H$_2$S completely reacts to COS at temperatures above 1000 °C.

$$H_2S + CO \longrightarrow COS + H_2 \qquad (7)$$

$$H_2S + CO_2 \longrightarrow COS + H_2O \qquad (8)$$

At the temperature of 1000 °C, 98.7% of the sulfur contained in the fuel was detected in the gas phase in the form of SO$_2$, COS and H$_2$S (Figure 3). The yield decreases with

increasing temperature. At 1300 °C only 2.3% of the sulfur contained in the fuel is released as $SO_2$, $H_2S$ and COS.

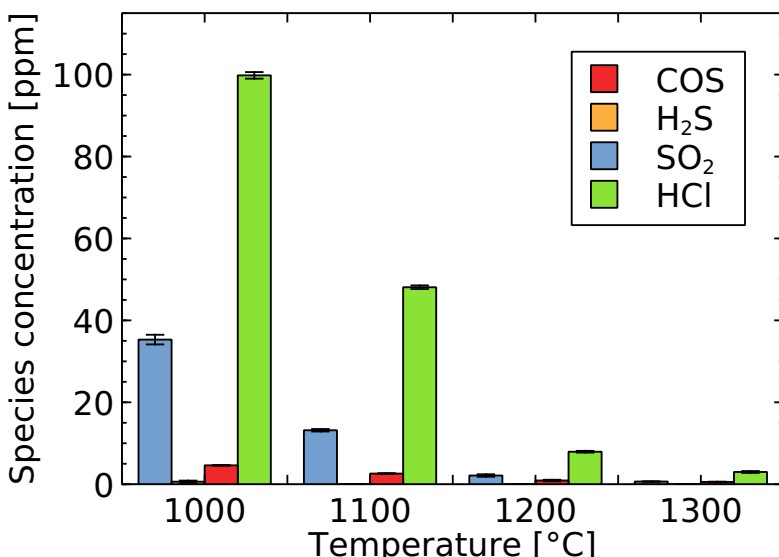

**Figure 2.** Concentrations in $N_2$ atmosphere.

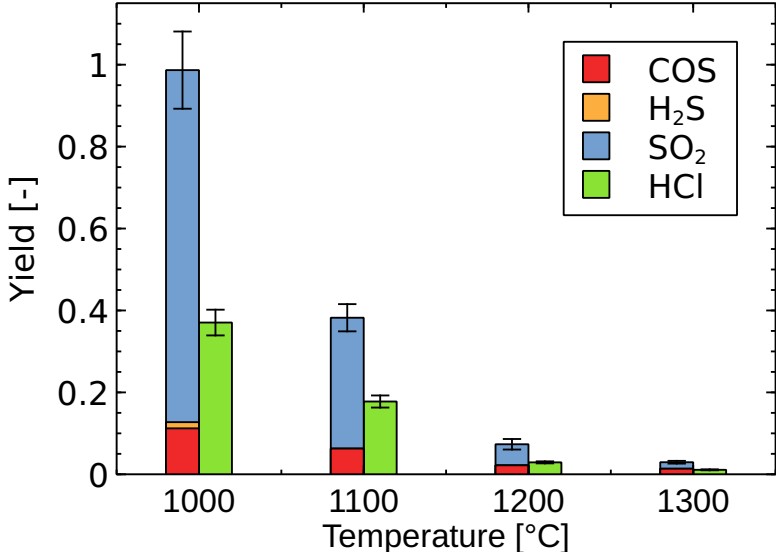

**Figure 3.** Yields of $SO_2$, COS, $H_2S$ and HCl in $N_2$ atmosphere.

Sulfur that is not released during pyrolysis could be released in other forms or remain in the char. Possible hydrocarbon sulfides in the gas phase could be carbon disulfide ($CS_2$) and methanethiol ($CH_4S$). The ionic currents of these species (76 and 47 amu) show a decreasing trend with increasing temperature in both atmospheres (Figures 4 and 5). This indicates that the sulfur reacts with the ash or is released in the gas phase in a different form.

One option is the formation of elementary sulfur under reducing conditions from $H_2S$ and $SO_2$ according to the Claus process (Equation (9)). Calculations in [39,40] show that with increasing temperature (1000 to 1300 °C) the conversion of $SO_2$ and $H_2S$ to elemental sulfur increases.

$$2\,H_2S + SO_2 \rightleftharpoons 3\,S + 2\,H_2O \tag{9}$$

Another possible reaction is the formation of calcium sulfide (CaS) according to Equations (10) and (11). Calcium oxide (CaO) reacts with COS or $H_2S$. The resulting CaS has a high melting point and remains in the ash. Studies show that this mechanism is enhanced with increasing temperature [41,42].

Figures 4 and 5 additionally show that the target ion currents of $CS_2$ are relatively similar in both atmospheres. However, the target ion currents of $CH_4S$ are about 100 times in the $CO_2$ atmosphere. This indicates that a $CO_2$ atmosphere promotes the release of $CH_4S$.

$$CaO + COS \rightleftharpoons CaS + CO_2 \tag{10}$$

$$CaO + H_2S \rightleftharpoons CaS + H_2O \tag{11}$$

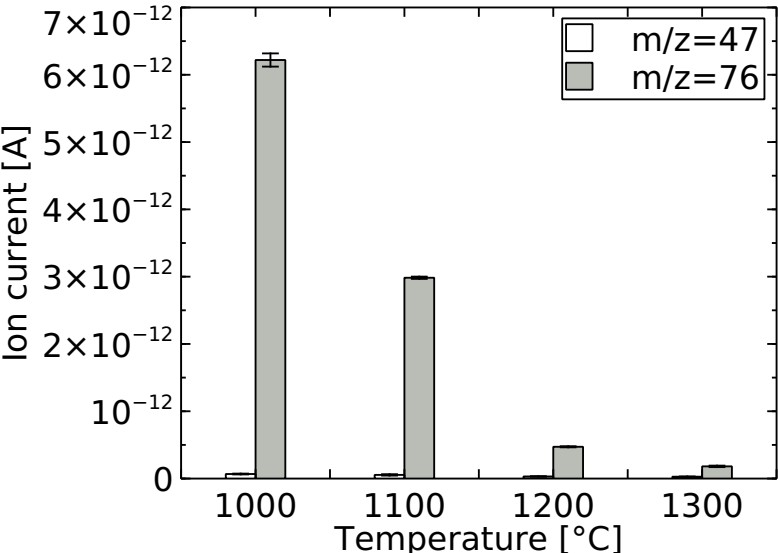

**Figure 4.** Ion currents in $N_2$ atmosphere.

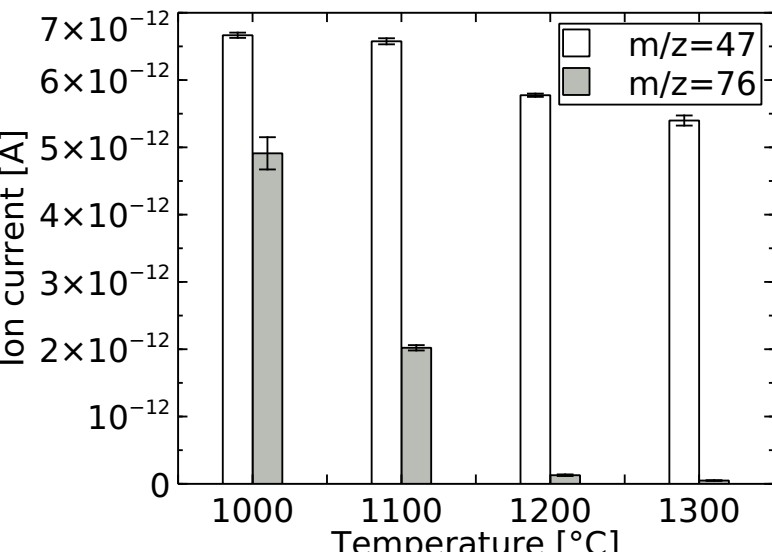

**Figure 5.** Ion currents in $CO_2$ atmosphere.

Figure 6 shows the concentration of sulfur compounds in $CO_2$ atmosphere. The concentrations of $SO_2$ at different temperatures are lower than in $N_2$ atmosphere. However, the COS concentrations are higher. $CO_2$ promotes the reaction of sulfur to COS according to Equation (8). Comparing the sulfur release in both atmospheres (Figures 3 and 7), it is clear that at temperatures below 1300 °C the sulfur release in both atmospheres is approximately the same. This shows that the sulfur contained in the fuel is already converted into the gas phase during pyrolysis, since the higher burnout during gasification has no influence on an increased sulfur release.

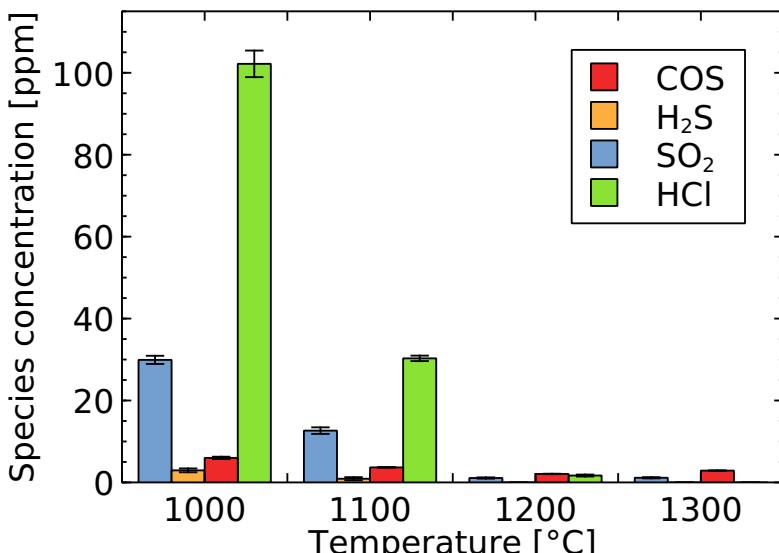

**Figure 6.** Concentrations in $CO_2$ atmosphere.

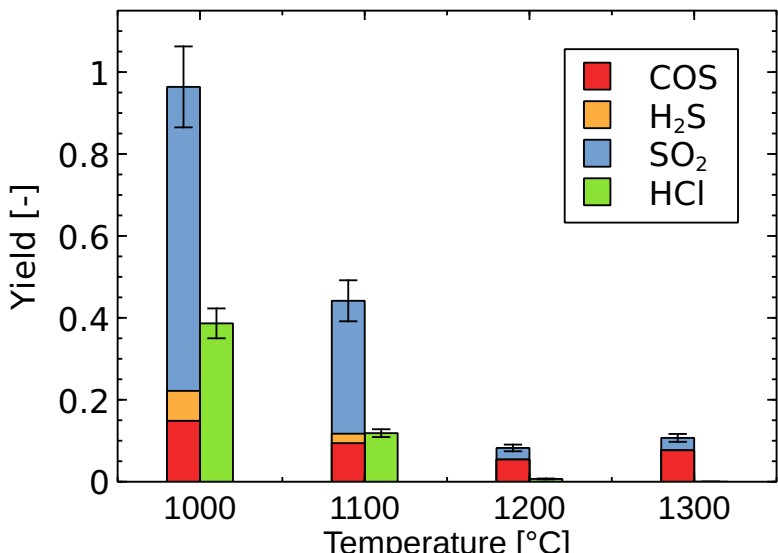

**Figure 7.** Yields of $SO_2$, COS, $H_2S$ and HCl in $CO_2$ atmosphere.

The concentrations of hydrogen chloride reach their maximum values at 1000 °C in both atmospheres (Figures 2 and 6). The change in atmosphere has no effect on the formation of HCl at this temperature. The formation decreases in both atmospheres with increasing temperature. A faster decrease is observed in carbon dioxide atmosphere. This could be due to the increased $CO_2$ concentration, which favors the release of chlorohydrocarbons such as chloromethane ($CH_3Cl$) and chloroethane ($C_2H_5Cl$), resulting in less HCl being formed. In [6], it is shown that with increasing temperature potassium and chlorine are released from biomass. The observation suggests that at high temperatures potassium reacts with chlorine to form KCl. Simulations performed in [7] show an increased release of chlorine as KCl in biomass with increasing temperature. In [43], it is shown that HCl reacts with calcium carbonate ($CaCO_3$) or iron oxide (FeO) to form calcium dichloride ($CaCl_2$) or iron dichloride ($FeCl_2$) (Equations (12) and (13)). The decrease in HCl concentration with increasing temperature is confirmed by these studies. Approximately 37.9% of the chlorine obtained in the fuel is released as HCl at 1000 °C in both environments (Figures 3 and 7).

$$2\,HCl + CaCO_3 \rightleftharpoons CaCl_2 + CO_2 + H_2O \qquad (12)$$

$$2\,HCl + FeO \rightleftharpoons FeCl_2 + H_2O \tag{13}$$

### 3.2. Combustion under Air and Oxy-Fuel Conditions

The experiments in $O_2/N_2$ and $O_2/CO_2$ atmospheres are carried out at two wall temperatures and four different stoichiometric conditions. The results of the sulfur and chlorine concentrations ($T$ = 1000 °C, 1300 °C) are shown in Figures 8–11. The yields are shown in Figures 12–15.

Figure 8 illustrates the concentrations in the gas phase of $SO_2$, COS and $H_2S$ at 1000 °C. In comparison with the pyrolysis experiments (Figure 2), all measured sulfur concentrations are lower, regardless of the stoichiometry during combustion. This is due to the fact that gaseous potassium compounds react with $SO_2$ and $O_2$ to form $K_2SO_4$ due to the higher oxygen content in the oxidizing atmosphere [44–46]. In all experiments, $H_2S$ is not detected.

The comparison of the results in air atmosphere (Figures 8 and 10) show a lower $SO_2$ concentration at 1000 °C than at 1300 °C. In addition to potassium, $SO_2$ can be bound by calcium oxide contained in the fuel according to Equations (14) and (15) [47,48]. However, at higher temperatures the reactions run in the opposite direction, so the sulfur remains in the gas phase as $SO_2$.

$$CaO + SO_2 + \frac{1}{2}\,O_2 \rightleftharpoons CaSO_4 \tag{14}$$

$$CaO + SO_2 \rightleftharpoons CaSO_3 \tag{15}$$

Figure 10 shows a decreasing $SO_2$ concentration with increasing stoichiometry. The $K_2SO_4$ formation can be promoted by the increasing oxygen partial pressure. Figures 8 and 10 further show the formation of COS only at sub-stoichiometric conditions ($\lambda < 1$). At an abundance of oxygen ($\lambda > 1$), COS is oxidized to $SO_2$. The sulfur yield in the gas phase at 1000 °C is nearly constant over stoichiometry and is about 2.5% (Figure 12) . At 1300 °C the yield amounts to 9.1% ($\lambda = 0.8$) and decreases to 2.7% ($\lambda = 1.1$) with increasing stoichiometry (Figure 14).

The comparison between the air and oxy-fuel environment at 1000 °C demonstrates higher $SO_2$ and COS concentrations during oxy-fuel combustion (Figures 8 and 9). A higher $CO_2$ partial pressure in $O_2/CO_2$ atmosphere promotes the formation of COS and inhibits the reaction of CaO with $SO_2$, since CaO is able to react with $CO_2$ to form $CaCO_3$ according to Equation (16). Thus, a higher temperature (1300 °C) has nearly no effect on the $SO_2$ concentrations in $O_2/CO_2$ atmosphere (Figure 11). The yield of sulfur in $O_2/CO_2$ atmosphere at is nearly constant over the stoichiometry (Figures 13 and 15). It amounts to 1000 °C, approximately 4.9%, and is two times higher as in air atmosphere.

$$CaO + CO_2 \rightleftharpoons CaCO_3 \tag{16}$$

While the $SO_2$ concentration at 1300 °C decreases with increasing stoichiometry in an air environment (Figure 10), no clear trend is observed in an oxy-fuel environment (Figure 11). The reason could be the formation of additional sulfur hydrocarbons, which are formed at substoichiometric conditions under oxy-fuel combustion. The COS concentration under oxy-fuel combustion at 1300 °C is higher than in air atmosphere as well and decreases with increasing stoichiometry (Figure 10). The maximum sulfur yield under oxy-fuel combustion amounts to 6.4% ($T$ = 1300 °C, $\lambda = 0.9$).

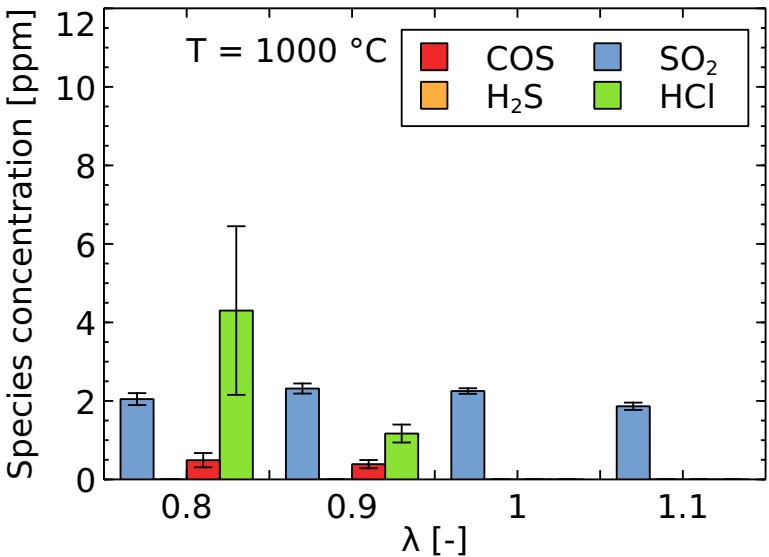

**Figure 8.** Concentrations in $O_2/N_2$ atmosphere.

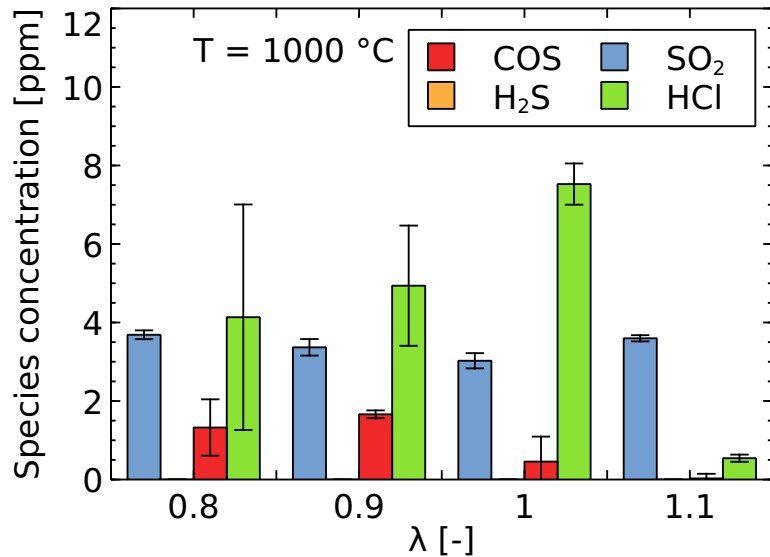

**Figure 9.** Concentrations in $O_2/CO_2$ atmosphere.

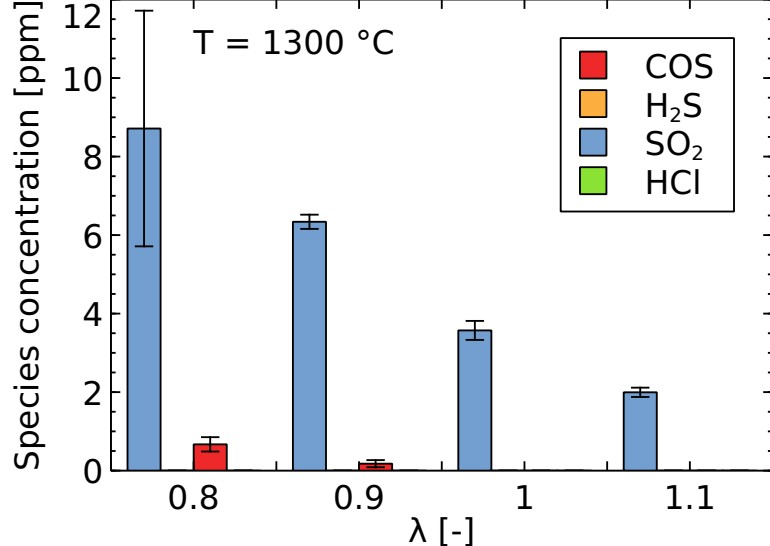

**Figure 10.** Concentrations in $O_2/N_2$ atmosphere.

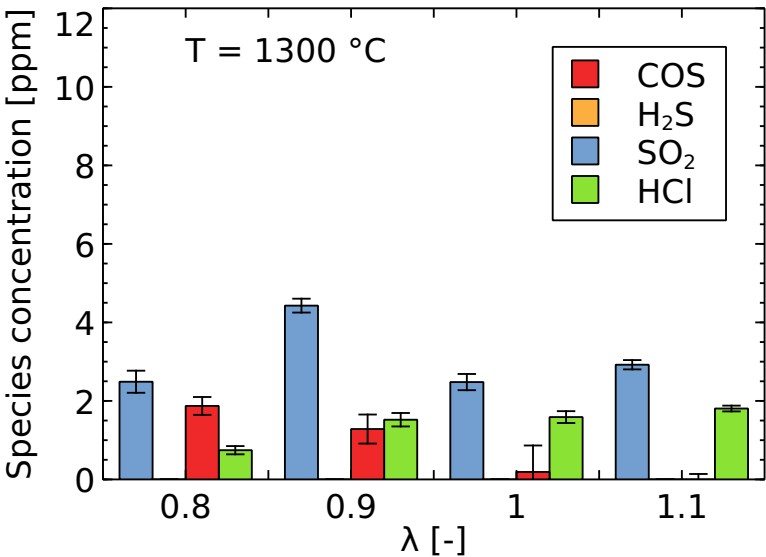

**Figure 11.** Concentrations in $O_2/CO_2$ atmosphere.

In addition, Figures 8–15 show the chloride concentrations and yields of the fuel as HCl. While in air atmosphere HCl is only detected at sub-stoichiometric conditions and at 1000 °C (Figures 8 and 10), HCl is detected in the oxy-fuel combustion at all experimental points (Figures 9 and 11). The $CO_2$ atmosphere can inhibit the formation of KCl by causing potassium to react with $CO_2$ to form potassium carbonate ($K_2CO_3$). As a result, more chlorine can be released as HCl. The release shows no clear trend over stoichiometry. This may be due to the release of chlorine as chlorohydrocarbons in sub-stoichiometric conditions. The thesis is supported by the fact that $K_2CO_3$ is found in fly ashes from biomass fired boilers [49].

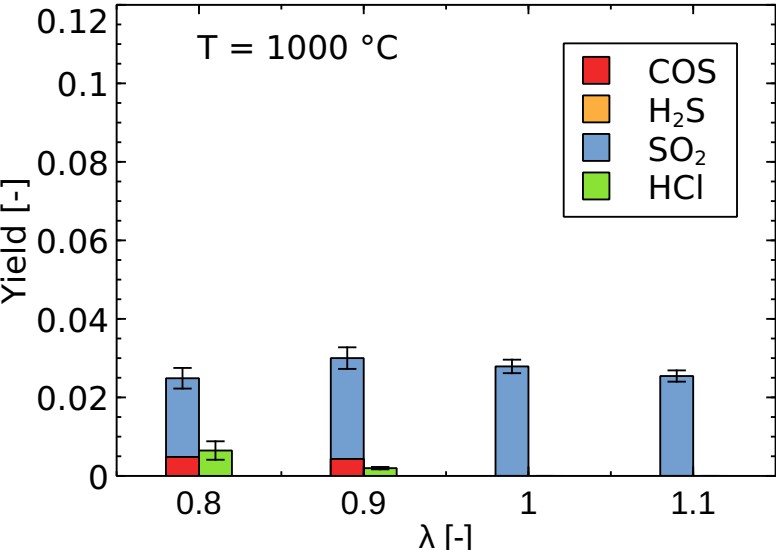

**Figure 12.** Yields of $SO_2$, COS, $H_2S$ and HCl in $O_2/N_2$ atmosphere.

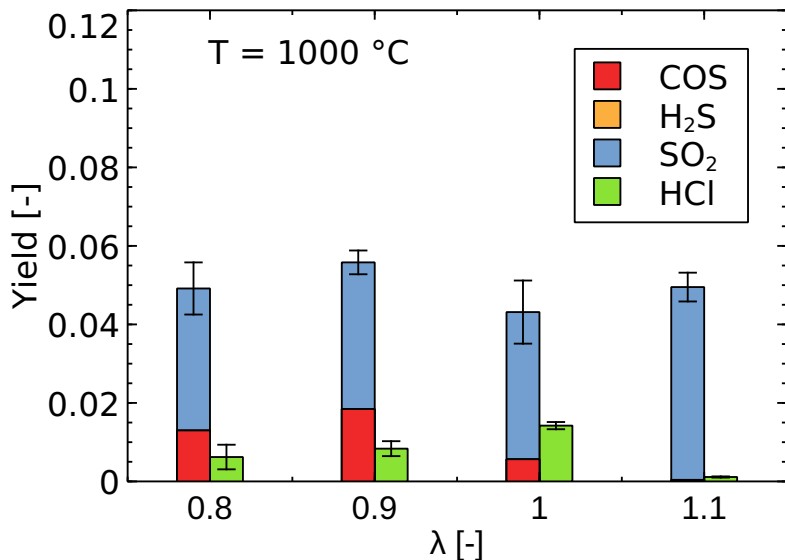

**Figure 13.** Yields of $SO_2$, COS, $H_2S$ and HCl in $O_2/CO_2$ atmosphere.

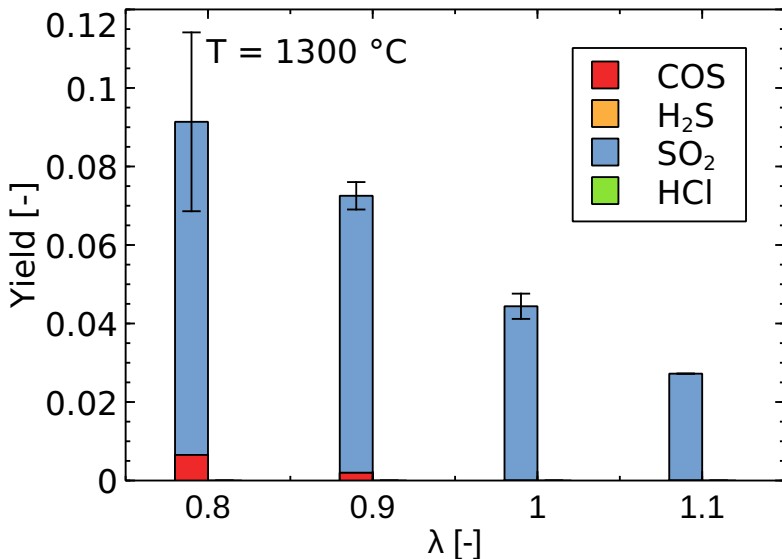

**Figure 14.** Yields of $SO_2$, COS, $H_2S$ and HCl in $O_2/N_2$ atmosphere.

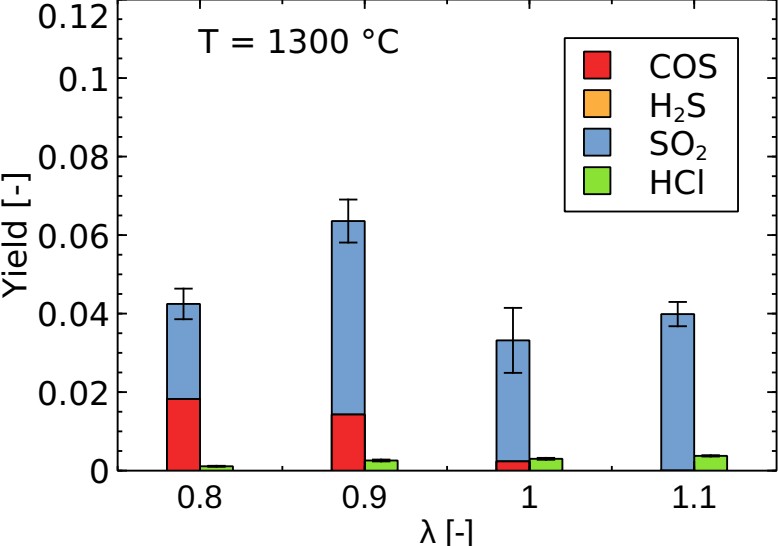

**Figure 15.** Yields of $SO_2$, COS, $H_2S$ and HCl in $O_2/CO_2$ atmosphere.

## 4. Conclusions

Decreasing concentrations of $SO_2$ and HCl with increasing temperature are observed in $N_2$ and $CO_2$ atmospheres. It is assumed elemental sulfur or calcium sulfide is formed. Ion currents of $CH_4S$ and $CS_2$ show a probable origin of this species. With increasing temperature, the currents are decreasing in both atmospheres. A decreasing HCl concentration with increasing temperature suggests more KCl is formed. This can result in corrosive deposits. Higher COS concentrations are detected in $CO_2$ atmosphere, which are decreasing with increasing temperature. COS could cause increased corrosion in an enriched $CO_2$ atmosphere. A total of 98.7% of fuel-bound sulfur volatilizes as $SO_2$, COS and $H_2S$ in $N_2$ atmosphere at 1000 °C. As HCl, 37.0% of the chlorine is released at this temperature in $N_2$ atmosphere. In $CO_2$ atmosphere, a similar total release of sulfur and chlorine is observed (1000 °C).

Combustion experiments show significantly lower concentrations of $SO_2$, COS, $H_2S$ and HCl as in $N_2$ and $CO_2$ atmospheres. It is assumed sulfur initially released during combustion and forms $K_2SO_4$ due to the oxidizing atmosphere. This promotes the formation of corrosive deposits. Higher $CO_2$ partial pressure can inhibit the formation of $K_2SO_4$ and promote the formation of $K_2CO_3$ at low temperatures, resulting in increased $SO_2$ and COS formation. No $H_2S$ was detected in the gas phase during any of the combustion trials.

Future work is needed to provide quantitative assessment of ash samples and the evaluation of sulfur and chlorinated hydrocarbons (such as $CH_4S$, $CS_2$, $CH_3Cl$ and $C_2H_5Cl$) in the early stage of combustion. In addition, low heating rate experiments are needed to obtain detailed information on the release kinetics.

**Author Contributions:** Conceptualization, C.Y. and M.R.; methodology, C.Y.; validation, C.Y., M.R. and J.S.; formal analysis, C.Y.; investigation, C.Y.; data curation, C.Y.; writing—original draft preparation, C.Y., M.R. and J.S.; writing—review and editing, C.Y. and M.R.; visualization, M.R.; supervision, J.S. and B.E.; funding acquisition, B.E. All authors have read and agreed to the published version of the manuscript.

**Funding:** This work was funded by the Deutsche Forschungsgemeinschaft (DFG, German Research Foundation)—215035359—SFB/TRR 129 'Oxyflame'.

**Data Availability Statement:** The experimental data will be published as soon as possible.

**Acknowledgments:** We acknowledge support by the Deutsche Forschungsgemeinschaft (DFG—German Research Foundation) and the Open Access Publishing Fund of Technical University of Darmstadt.

**Conflicts of Interest:** The authors declare no conflict of interest.

## Abbreviations

The following abbreviations are used in this manuscript:

Roman

| | |
|---|---|
| $Al_2O_3$ | Aluminium oxide |
| CaO | Calciumoxid |
| $CaCO_3$ | Calcium carbonate |
| $CaCl_2$ | Calcium dichloride |
| CaS | Calcium sulfide |
| Cl | Sodium |
| $CH_3Cl$ | Chloromethane |
| $CH_4S$ | Methanethiol |
| $C_2H_5Cl$ | Chloroethane |
| $CO_2$ | Carbon dioxide |
| COS | Carbonyl sulfide |
| $CaSO_3$ | Calcium sulfite |
| $CS_2$ | Carbon disulfide |
| $CaSO_4$ | Calcium sulfate |
| FeO | Iron oxide |

| | |
|---|---|
| $FeCl_2$ | Iron dichloride |
| $H_2$ | Hydrogen |
| $H_2O$ | Water |
| $H_2S$ | Hydrogen sulfide |
| HCl | Hydrogen chloride |
| K | Potassium |
| KCl | Potassium chloride |
| $K_2CO_3$ | Dipotassium carbonate |
| $K_2O$ | Dipotassium oxide |
| $K_2SO_4$ | Dipotassium sulfate |
| $\dot{m}_{Fuel}$ | Fuel mass flow ($kg\,s^{-1}$) |
| MgO | Magnesium oxide |
| $\dot{n}$ | Molar mass flow rate ($mol\,s^{-1}$) |
| $N_2$ | Nitrogen |
| NaCl | Sodium chloride |
| $O_2$ | Oxygen |
| $p$ | Pressure (bar) |
| $R$ | Universal gas constant ($8.314\,J\,mol^{-1}\,K^{-1}$) |
| S | Sulfur |
| $SO_2$ | Sulfur dioxide |
| $SiO_2$ | Quartz |
| $T$ | Temperature (°C) |
| $u_c$ | Combined standard uncertainty |
| $u_i$ | Uncertainty of the individual measurands |
| $\dot{V}$ | Volume flow ($m^3\,s^{-1}$) |
| $w_{Cl}$ | Chlorine content of the fuel (%) |
| $w_S$ | Sulfur content of the fuel (%) |
| $x$ | Volume fraction (%) |
| $y$ | Yield (-) |

Greek

| | |
|---|---|
| $\sigma$ | Standard deviation |
| $\lambda$ | Stoichiometric ratio (-) |

Abbreviations

| | |
|---|---|
| amu | Atomic mass unit |
| BECCUS | Bioenergy carbon capture, utilization and storage |
| CCUS | Carbon capture, utilization and storage |
| EFR | Entrained flow reactor |
| FG | Flue gas |
| MV | Measured value |
| MEA | Monoethanolamine scrubbing |
| m/z | Mass to charge ratio |
| WNS | Walnut shell |
| PTFE | Polytetrafluoroethylene |

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
