# Peer review of "Release of Sulfur and Chlorine Gas Species during Combustion and Pyrolysis of Walnut Shells in an Entrained Flow Reactor"

_energies, doi:10.3390/en16155684_

Round 1

Reviewer 1 Report

Dear Authors,

Many thanks for your submitted work. It was interesting, However, not that novel. Some of definitions are a bit difficult to understand and cope with! First of all if you consider Walnut as biomass there are plenty works that have similar idea and they found trace gases or spices for special applications. Like the works recently done in Sweden and other EU countries according to horizon 2020 for substitution of NG with gassified biomass. Please find below references or titles that follow the same idea for the end users. You need to carefully mention them in your introduction.

10.1016/j.energy.2014.08.092

10.1021/acs.energyfuels.8b01232

"The effect of impurities on syngas combustion " 

1- I am not sure that Walnut was not considered! Please be very careful about that. 

2- In my opinion your English needs more careful attention: Line 83 and 84 in page 2 is a bit problematic and not easy to understand in term of English as well. 

3- All abbreviations or chemical formula must be defined prior to their first use. Please be specific! Like COS in abstract what is it stands for!!!!

4- From your abstract and title I couldnt follow you are sticking to combustion or pyrolysis? It couldnt be both as your setup at least doesnt represent such idea! As your setup and its name shows gasification! Why is so? Maybe I am not following your idea!!

5- I couldnt understand oxy-fuel atmosphere! We have Oxyfuel combustion not as atmosphere! as you have nitrogen and etc. as single medium! why it is complex? What is fuel in oxyfuel? Is it walnut? So the atmosphere is pure oxygen?

6- You should provide more details and conditions regarding your reactor! 

7- Please provide the proximate analysis of your Walnut!

8- What is the application of such work? Could it be used in industrial case? Do we have such amounts of Walnut? 

9- What about nitrogen compounds! Your temperature ranges are quite high! So you have NOx emission as well!  

In my opinion this work is not that novel! Some definitions are also difficult to understand for the experts in this topic

Reviewer 2 Report

The paper shows results from thermochemical treatment of walnut shells. Experiments were carried out in N2, CO2, air and oxy-fuel atmosphere at different temperatures. The release of sulphur and chlorine were studied. The research shown in the manuscript is basic, but can probably be published in Energies.

Comments:

1) Only in the introduction, the authors cited 49 papers to support what is written, and 71 references were used to support the whole work. Sentences such as in lines 72-74: “Chlorine which is released at low temperatures in the form of HCl [23,24,35,3840] can potentially be recovered by secondary reactions with available metals in the fuel [23,41,42].” This is common knowledge, nine citations to support this is not needed.

Try to avoid the usage of clumped references and reduce the amount. For a research paper around 40-50 references are more than enough.

2) Table 1 the elemental composition of the walnut shell ash.

The usage of oxide form in the elemental ash composition is misleading because you actually don’t have the oxides in the biomass. The usage of oxides in elemental composition comes from old hard coal combustion papers and people still use it as a convention but it doesn’t make sense.

 Apart from this, it also complicates the comparison, e.g., for P2O5 you have 2P and 5O, for CaO you have 1Ca and 1O, for SiO2 you have 1Si and 2O, and so on. So when using the oxide form of the elements you can’t compare, e.g., you have 2.2% MgO and 2.2% Na2O (i.e., the same amount for both), and the amount of each element, in reality, are not the same.

The authors could recalculate the elemental composition to show also the percentage of each element instead of the oxides.

3) The conclusion section should only contain a brief summary of the main findings.
